# The Graphical Studies of the Major Molecular Interactions for Neural Cell Adhesion Molecule (NCAM) Polysialylation by Incorporating Wenxiang Diagram into NMR Spectroscopy

**DOI:** 10.3390/ijms232315128

**Published:** 2022-12-01

**Authors:** Guo-Ping Zhou, Ri-Bo Huang

**Affiliations:** 1National Engineering Research Center for Non-Food Biorefinery, Guangxi Academy of Sciences, Nanning 530004, China; 2Rocky Mount Life Sciences Institute, Rocky Mount, NC 27804, USA; 3College of Life Science and Technology, Guangxi University, Nanning 530004, China; 4State Key Laboratory for Conservation and Utilization of Subtropical Agro-Bioresources, Guangxi University, Nanning 530004, China

**Keywords:** NMR, wenxiang diagram, sialic acid (Sia), polysialic acid (polySia), polysialyltransferases (polyST), ST8Sia II (STX), ST8Sia IV (PST), neural cell adhesion molecule (NCAM), polysialyltransferase domain (PSTD), polybasic region (PBR)

## Abstract

Polysialylation is a process of polysialic acid (polySia) addition to neural cell adhesion molecule (NCAM), which is associated with tumor cell migration and progression in many metastatic cancers and neurocognition. Polysialylation can be catalyzed by two highly homologous mammalian polysialyltransferases (polySTs), ST8Sia II (STX) and ST8Sia IV (PST). It has been proposed that two polybasic domains, polybasic region (PBR) and polysialyltransferase domain (PSTD) in polySTs, are possible binding sites for the intermolecular interactions of polyST–NCAM and polyST–polySia, respectively, as well as the intramolecular interaction of PSTD–PBR. In this study, Chou’s wenxiang diagrams of the PSTD and PBR are used to determine the key amino acids of these intermolecular and intramolecular interactions, and thus it may be helpful for the identification of the crucial amino acids in the polyST and for the understanding of the molecular mechanism of NCAM polysialylation by incorporating the wenxiang diagram and molecular modeling into NMR spectroscopy.

## 1. Introduction

The previous studies have proposed that polysialylation of neural cell adhesion molecule (NCAM) is associated with tumor cell migration and progression in many metastatic cancers and neurocognition [1,2,3,4,5]. Polysialylation is a process of polySia addition to NCAM, which plays a crucial role in neural development, neural regeneration and plastic processes in the vertebrate brain associated with neurite outgrowth, axonal pathfinding and learning and memory [6,7,8,9].

It has been known that two highly homologous mammalian polysialyltransferases (polySTs), ST8Sia II (STX) and ST8Sia IV (PST), can catalyze NCAM polysialylation [10,11,12,13], and two polybasic domains, polybasic region (PBR) and polysialyltransferase domain (PSTD) in polySTs, play key roles in affecting polyST activity or NCAM polysialylation [14,15].

PSTD has been identified as a polybasic motif of 32 amino acids unique in 2006 [15]. Based on the results of the in vitro experimental data and molecular modeling analysis [16,17,18,19,20], the hypothesis of the interaction between CMP-Sia and PSTD was proposed in the previous studies and further supported by the biophysical experiments [21,22].

Similar to the PSTD, PBR is conserved in PST and STX and is located equidistant between the large sialyl motif (SML) and the transmembrane domain (TMD) in each enzyme [14,15]. It consists of 35 amino acids, of which 7 are basic amino acids, such as Arg and Lys residues [14]. Calley et al. reported another interaction between the PBR and NCAM by the NMR experiments [23]. It is interesting that a possible cooperative effect between PBR and PSTD within tumor-target polyST was described [18].

The introduction of various graphic methods into biological systems has been successfully used to analyze many important biological problems, such as enzyme-catalyzed reactions [24,25,26], slow conformational change [27], protein folding kinetics and folding rates [28,29], inhibition of HIV-1 reverse transcriptase [30,31,32], nonsteady drug metabolism systems [33], hepatitis B virus (HBV) infections [34], HBV virus gene missense mutation [35] and evolution of biological sequences [36]. In addition, recently, Chou’s wenxiang diagram [37,38,39,40] has been used to study protein coil-coiled interactions [41,42], prediction of alpha-helical protein stability and misfolding mechanism of the protein [43,44,45]. 

Inspiration for generating Chou’s wenxiang diagram is related to the Chinese wenxiang, a coil-like incense widely used in China’s countryside to repel mosquitoes [37,38]. It is amazing that some important features in an α-helix of a protein can be visualized using a novel 2D diagram and looks like the Chinese wenxiang (also see http://doi.org/10.4236/ns.2011.310111) for a brief summary of the advantages of using the wenxiang diagrams to represent helices in proteins). There are three major advantages of an α-helix structural wenxiang diagram in proteins: (1) each amino acid relative location on the helical structure could be clearly shown on a wenxiang diagram no matter how long this helical length; (2) hydrophobic and hydrophilic residues could be distinctly distributed at different regions of the wenxiang diagram for all amphiphilic helical proteins; (3) both N- and C-terminal residues of an α-helix could be displayed on a wenxiang diagram [37,38,39,40,41,42,43].

Recent studies have discovered that the molecular mechanism of NCAM polysialylation is related to the intramolecular and intermolecular interactions between the PSTD and polySia and the PSTD and PBR based on the NMR spectroscopy and molecular modeling analyses [19,21]. In order to determine the precise molecular details of both the intramolecular and intermolecular interactions required for polysialylation, we incorporated Chou’s wenxiang diagram into NMR spectroscopy in the current study.

Based on the previous investigations, there are the major helices in both the PSTD and PBR domains of polyST molecule, and their ranges are S257–N271 and S75–L89, respectively [14,16,19]. In this study, wenxiang diagrams of the PSTD and PBR are used to analyze possible intermolecular and intramolecular interactions for NCAM polysialylation by incorporating NMR spectroscopy. The identification of the crucial amino acids in the PSTD of ST8Sia IV may be helpful to inhibit NCAM polysialylation and tumor cell migrations.

## 2. Results

### 2.1. The Three Key Amino Acids of the PSTD–Polysia Interaction Are Predicted by the PSTD Wenxiang Diagram

The 3D structure of the PSTD (K246-R277) in ST8Sia IV contains a long helix H2 from S257 to N271 (Figure 1b) based on the previous investigations [16,17]. In the previous studies, most crucial residues affecting the PSTD–polyST interaction are located in the helical middle and C-terminate of the PSTD (Figure 1). The depositions of these 15 residues of the H2 helix on the wenxiang diagram are shown in Figure 2a, where all three basic residues, R259, H262 and R265, are located in the semicircular area of the figure (polar residual region), and most hydrophobic residues are concentrated in another semicircular area of the figure (nonpolar residual region). 

According to the intuitive observation, the distances between polySia and the three hydrophobic residues, V264, Y267 and W268, are shorter than the polySia and the other residues in the wenxiang diagram (Figure 2a). These three residues are located in the middle of the helical middle (V264) and the C-terminate of the PSTD’s helix (Y267 and W268), respectively. Although L269 is located in another half of the wenxiang diagram (hydrophilic region), it is a hydrophobic residue located between Y267 and W268 in the PSTD wenxiang diagram and is closed to Y267 and W268. Thus, it is possible that V264, Y267, W268 and L269 could be predicted to be the key amino acids for the PSTD–polySia interaction. Actually, the previous NMR data have shown that these four residues are crucial amino acids for the interaction between the PSTD and polySia [19,21]. This indicates that the findings from both the wenxiang diagram and the NMR experiments are consistent.

### 2.2. Characteristics of the Helical Wenxiang Diagrams in the PBR

The 3D structure of the PBR in ST8Sia IV contains a helix ranging from S75 to L89. According to the 3D model of the ST8Sia 4 molecule, the orientation of both the PBR and the PSTD spirals is reverse, and the spirals are close to each other within the ST8Sia molecule, which is to say that the C-terminus of the PBR is close to the N-terminus of the PSTD (Figure 1b).

Thus, the C-terminal residues in PBR’s wenxiang diagram should be on the outermost side of the spiral, and N-terminal residues should be on the innermost side of the spiral (Figure 2b), which is consistent with the conformational orientation of PSTD and PBR within ST8Sia IV (Figure 1a). In addition, similar to the wenxiang diagram of the helix H2 in the PSTD, most nonpolar residues in the PBR’s helix are also located in the semicircular side of the wenxiang diagram, and most polar residues, including all three polybasic residues, R82, K83 and R87, are concentrated in other half side of the wenxiang diagram (Figure 2b). 

### 2.3. The Graphical Analysis of the PBR–FN1 Interaction 

Bhide et al. [23] have probed the interaction between the PBR and NCAM’s first fibronectin type III repeat (FN1) using isothermal titration calorimetry and nuclear magnetic resonance (NMR) spectroscopy. They observed direct and specific binding between FN1 and the PBR peptide that is dependent upon acidic residues in FN1 and basic residues of the PBR and found that R82 and R93 are key residues for NCAM recognition and polysialylation [23].

According to 3D model of ST8Sia IV, R93 is outside of the PBR’s helix and is flanked by negatively charged glutamate and aspartate residues (E92-R93-D94) in ST8Sia IV, and there is a random coil region between this E92–R93–D94 loop and the PBR’s helix. In order to display a significant interaction between the PBR and FN1, the positions of R93, R82 and K83 should be closed to FN1 molecule in the space, and they are all exposed on the surface of ST8Sia IV model and are located in same side of the surface model of ST8Sia IV (Figure 1c). These characteristics are be reflected in Figure 2c, where R82 and K83 in the PBR wenxiang diagram are close to R93. Incorporating the PBR wenxiang diagram into the molecular modeling of ST8sia IV may intuitively explain why R82, K83 and R93 are key amino acids for the PBR–FN1 interaction, and it also supports the previous findings using NMR spectroscopy [23].

### 2.4. The PBR–PTSD Interaction Are Characterized by Their Combined Wenxiang Diagrams

In order to characterize the intramolecular interaction between the PBR and the PSTD, the relative positions of these two wenxiang diagrams should be determined. The three hydrophobic residues, V264, Y267 and W268, from the PSTD wenxiang diagram have been contributed to the intermolecular interaction between the PSTD and polySia (Figure 2), and according to the molecular model, the middle part of the PSTD helix is close to the C-terminus of the PBR helix (Figure 1). Thus, the related position of the PBR to the PSTD should be displayed as Figure 3. In this combined wenxiang diagram, the two shortest distances between the PSTD–PBR should be displayed among two pairs of residues, F88 (PBR) and Y267 (PSTD), and L89 (PBR) and A263 (PSTD). Their distances are 5.8 Å and are 5.6 Å, respectively (Figure 1). Moreover, 5.6–5.8 Å belong to a weak restraint distance range for interactions between molecules or domains. This weak intramolecular interaction between the PBR and PSTD domains is proposed based on the previous modeling analysis [18,19]. Thus, the relative position of the PBR’s helix and the PSTD’s helix in the combined wenxiang diagram is consistent with that of the PBR to the PSTD in the molecular model. 

## 3. Discussion

In this study, the wenxiang diagrams of the PSTD and PBR were produced according to the relative positions of their helices in molecular modeling (Figure 1). When the polySia molecule is located in the position as shown in the PSTD wenxiang diagram (Figure 2a), the four key amino acids, V264, Y267, W268 and L269, for the PSTD–polySia interaction could be proposed to be incorporated into the molecular modeling conformation.

According to the previous NMR results, the chemical shift changes for PSTD–polySia interaction could be detected in most nonpolar residues. However, the most significant changes in chemical shift or chemical shift perturbation (CSP) were found to be V264, Y267, W268 and L269 [18,19]. These findings supported that the characteristic distribution of the amino acids in the PSTD wenxiang diagram and the relative position of polySia molecule to the wenxiang diagram (Figure 2a) are reasonable, and the predicted key amino acids for PSTD–polySia interaction are consistent with the NMR data [18,19]. 

The three crucial amino acids, R82, K83 and R93, in the PBR are verified for the PBR–FN1 interaction by the relative positions of the wenxiang diagram of the PBR and FN1 in Figure 4. Figure 4 is actually a combination of Figure 2a,c. Although R93 is outside of the PBR helix, it should be on the same side as R82 and K93, and these three positive-charged residues have the shortest distances to the FN1 molecule. Thus, the relative positions of the three residues and FN1 are reasonably shown in Figure 2c. Obviously, the intermolecular interaction between the PBR and FN1 should be an electrostatic interaction. Comparing with an intermolecular polySia–PSTD interaction, the PBR–FN1 one is a relatively strong interaction.

F88 and L89 in the PBR and A263 and Y267 in the PSTD are crucial residues for the PBR–PSTD intramolecular interaction. This finding from ST8Sia IV molecular modeling was supported by the combined wenxiang diagram of the PBR and the PSTD (Figure 3). Thus, these results indicate that the observational results using wenxiang diagrams are consistent with those from experimental and molecular models [18,19]. 

The above three interactions (polySia–PSTD, FN1–PBR and PSTD–PBR interactions), may participate cooperatively in the NCAM polysialylation process [18,19]. In addition to these intermolecular and intramolecular interactions, the PSTD might also interact with CMP-Sia in the Golgi apparatus before polySia formation, according to the previous NMR studies [21]. Comparing HSQC spectra of the (CMP-Sia)–PSTD and polySia–PSTD interactions, the binding sites of CMP-Sia are mostly bound to the outside of the long helix of PSTD, and the binding sites of polySia are mostly bound to the long helix of the PSTD. Another major difference between the (CMP-Sia)–PSTD and poly–PSTD interactions is that the faster exchange is displayed in the HSQC spectra of the (CMP-Sia)–PSTD interaction, and a slow exchange is observed in the HSQC spectra of the polySia–PSTD interaction. Thus, it is possible that CMP-Sia is fist bound to the PSTD in ST8Sia IV to generate polySia, and the exchange would slow down until the formation of a polySia chain [18]. Thus, the previous studies have suggested that the PSTD may play bifunctional roles: on the one hand, the PSTD may catalyze the polySia formation through its interaction with CMP-Sia, and on the other hand, it may also be a carrier to transfer the formed polySia chains to NCAM during the polysialylation process [18,19]. Therefore, (CMP-Sia)-PSTD interaction should be also involved in the cooperative interaction system for NCAM polysialylation [21].

Based on these findings, a possible molecular mechanism of NCAM polysialylation and cell migration was proposed in the previous study: In the beginning, two adjacent cells are usually connected through NCAM molecules located in the cell surfaces. In order to initiate NCAM polysialylation, CMP-Sia, which is transported into the lumen of the Golgi complex, is fist bound to the PSTD in ST8Sia IV to generate polySia; then, formed polySia chains are released from the PSTD due to its configuration change when the PBR is bound to FN1 in NCAM; and finally, two connected cells are separated, and cell migration occurs due to the repulsive effect of the negative-charged anions between polySia chains located in the NCAM [18]. This model provides a basic pathway from NCAM polysialylation and cell migration occurrence. Obviously, the current schematic diagram based on wenxiang diagrams are also nicely consistent with the previous NCAM polysialylation and cell migration model. 

## 4. Materials and Methods

### 4.1. The Sequences of the Helical Domains in the PBR and the PSTD within ST8Sia IV Molecule

According to the ST8Sia IV model [14,15,16,17,18], the sequence of helical domain in the PBR (residues 71–105) is:

S^75^SLVLEIRKNILRFL^89^, and the sequence of the helical domain in the PSTD (residues 246–277) within the ST8Sia IV molecule is:

S^257^LRLIHAVRGYWLTN^271^.

### 4.2. Ensemble Principle of Wenxiang Diagram of an α-Helix 

It has been clearly known that in amphiphilic helical proteins, most hydrophobic residues are distinctly distributed in one half of each wenxiang diagram, while most hydrophilic residues are distributed in the other half [37,41] according to the wenxiang diagram coordinate system [38].

Thus, a wenxiang diagram of an α-helix can be viewed as a two-dimensional diagram generated by its conical projection [37,38,40,41]. The projected image of a helix could be viewed as a planar spiral with a continuously varying radius. If a helical COOH-terminal is close to its projected plane and NH_2_-terminal is far away the plane, then the COOH-terminal of the helix lies near the center of the projected plane and the NH_2_-terminal lies at the outer rim of the diagram. 

However, if two helices are antiparallel, one of their combined wenxiang diagrams should display that its NH_2_-terminal lies near the center of the projected plane and that the COOH-terminal of the helix lies near the center of the projected plane; and for another helical wenxiang diagram, its NH_2_-terminal should be far away from the center of the projected plane.

## 5. Conclusions

In a traditional helical wheel [38], the information provided regarding α-helices of a protein and their lengths is very limited. In contrast, wenxiang diagrams can be used to represent α-helixes regardless of how long they are, and they provide much more information about the physicochemical features of the constituent amino acids and their distribution features in this 2D diagram. Furthermore, when incorporated with NMR, the molecular model of proteins’ 3D structure, as well as other biological/biophysical experimental results, the wenxiang diagram can be applied for analyzing stability of α-helical protein and identifying or verifying the key residues that play the most important role for the intramolecular and intermolecular interactions between two neighboring helix domains such as the PSTD and PBR, the PSTD and polySia, and PBR and FN1. In summary, our combined wenxiang diagrams of the PSTD–PBR interaction predict, for the first time, the PSTD–PBR intramolecular interaction and the crucial amino acids from these two domains. These predictions should be reasonable due to the intuitive and clear logic display of the integrated graphic system and have been supported by recent molecular modeling analysis. We found that two intermolecular and one intramolecular interaction could be perfectly integrated in a graphic system, and each interaction is able to be affected by another interaction. As shown in Figure 4, you can imagine that the PBR’s conformation should be changed when the PBR is bound to FN1, inducing a conformational change of the PSTD resulting in the polySia’s release from the bound PSTD, and finally resulting in the free polySia being bound to NCAM. Thus, this novel graphic system helps us to deeply understand the NCAM polysialylation mechanism, as well as the drug design of inhibiting tumor cell migration.

## Figures and Tables

**Figure 1 ijms-23-15128-f001:**
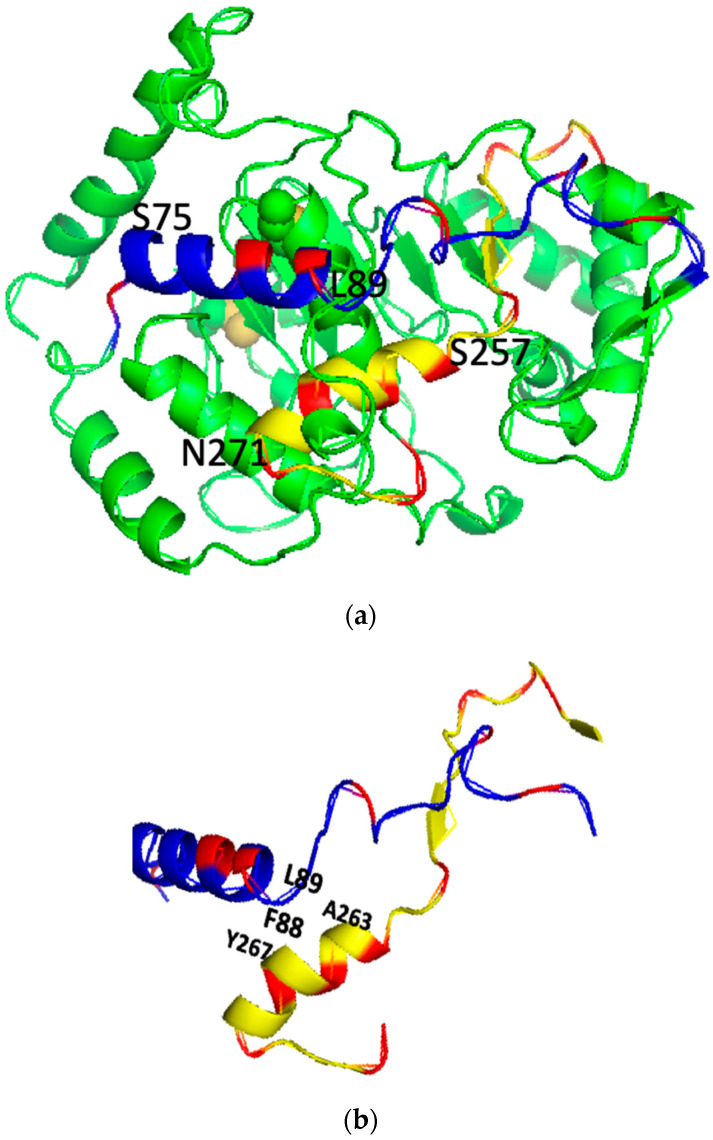
The predicted 3D backbone model of human ST8SiaIV was generated using Phyre2 sever [46] (**a**), the extracted PBR (blue) and PSTD (yellow) backbone motifs from the model (**a**) is shown in (**b**) and the surface model of the ST8Sia IV is displayed in (**c**). All basic residues are labeled in red. Above models were displayed using PyMol 2.5.

**Figure 2 ijms-23-15128-f002:**
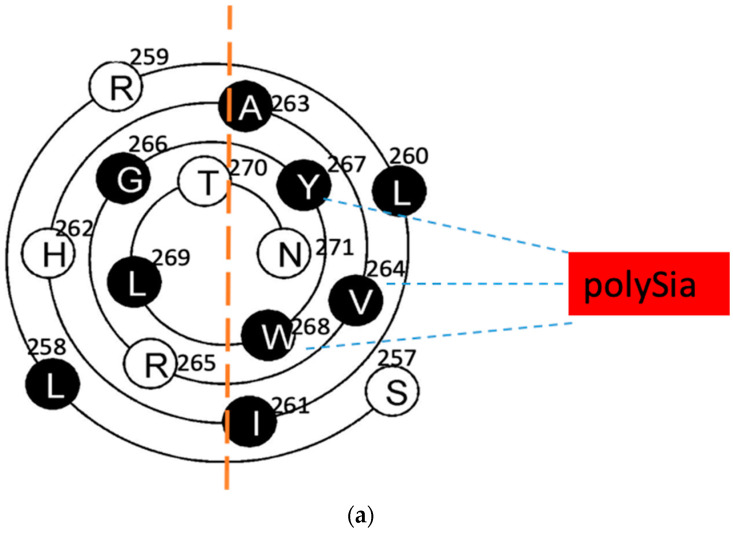
The wenxiang diagrams of the long helices in the PSTD (**a**) and the PBR (**b**), the schematics of the interactions between the PSTD and polySia (**a**) and the PBR and the FN1 in NCAM (**c**) using the helical wenxiang diagrams of the PSTD and the PBR, respectively. All hydrophobic residues in the wenxiang diagrams are labeled in black; the closest distances between the three hydrophobic residues (V264, Y267, W268) and polySia, and between the three basic residues (R82, K87, R93) and FN1, are labeled in blue dash and red dash lines, respectively.

**Figure 3 ijms-23-15128-f003:**
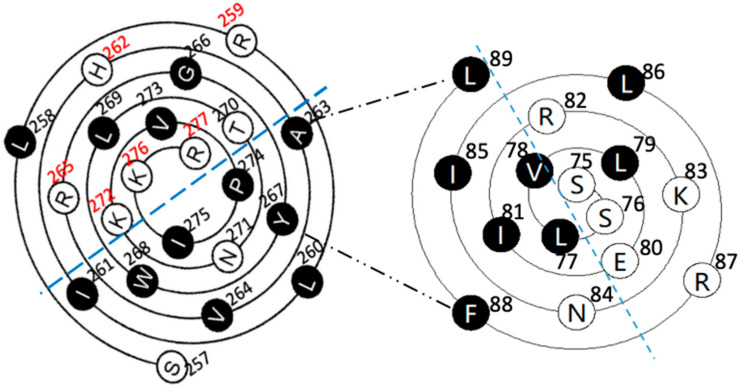
The schematic of the relative position and interaction between the PSTD and the PBR using the helical wenxiang diagrams of the PSTD and the PBR, respectively. All hydrophobic residues are labelled in black, and the closest distances between the residues of the two helices are in dash-dotted lines.

**Figure 4 ijms-23-15128-f004:**
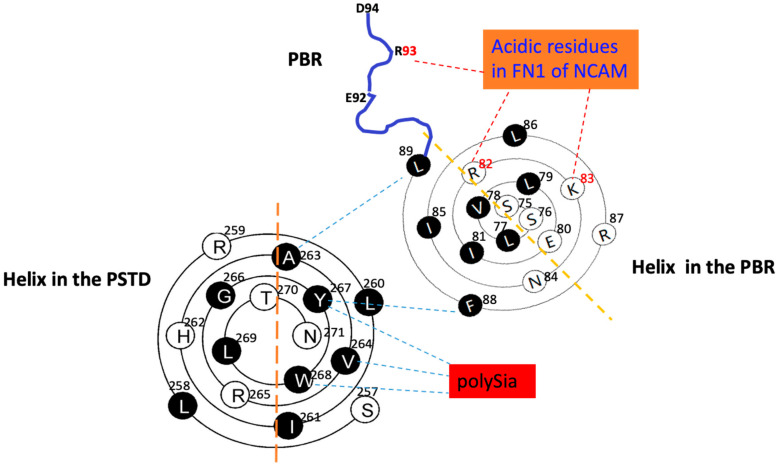
The system schematic of the relative position and interactions of the PSTD-polySia, PBR-FN1 and the PSTD-PBR using the helical wenxiang diagrams of the PSTD and the PBR, respectively. Actually, it is a combination of Figure 2a,c.

## Data Availability

Not applicable.

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
