# Peer review of "The Graphical Studies of the Major Molecular Interactions for Neural Cell Adhesion Molecule (NCAM) Polysialylation by Incorporating Wenxiang Diagram into NMR Spectroscopy"

_ijms, 2022, doi:10.3390/ijms232315128_

Round 1

Reviewer 1 Report

The manuscript titled “The Graphical Studies of the Major Molecular Interactions for NCAM Polysialylation by Incorporating Wenxiang Diagram into the NMR Spectroscopy” by Zhou and Huang reported the use of Chou’s wenxiang diagrams of the PSTD and PBR for determining the intermolecular and intramolecular interactions in NMR Spectroscopy. This approach may be useful in finding key amino acids of intermolecular and intramolecular interactions of proteins, peptides or the modified. The manuscript was well written and organized, and the experiments were performed properly and the conclusions were justified by the data.

Minors:

1.        NCAM in the title may be replaced by its non-abbreviated name.

2.        Refs: Be consistent for writing the authors’ names.  e.g.  ref 18. Zhou GP, …  to Zhou G. P. ?; ref 21. Liao SM, … to Liao S. M.

3.        Figures: More detailed figure legends may be provided, e.g. how the images were generated, what were the residues with black background, etc.

Author Response

Comments and Suggestions for Authors

Reviewer 1:

The manuscript titled “The Graphical Studies of the Major Molecular Interactions for NCAM Polysialylation by Incorporating Wenxiang Diagram into the NMR Spectroscopy” by Zhou and Huang reported the use of Chou’s wenxiang diagrams of the PSTD and PBR for determining the intermolecular and intramolecular interactions in NMR Spectroscopy. This approach may be useful in finding key amino acids of intermolecular and intramolecular interactions of proteins, peptides or the modified. The manuscript was well written and organized, and the experiments were performed properly and the conclusions were justified by the data.

Response of Authors:

Many thanks to Reviwer1’s very valuable comments.

Minors:

  1. NCAM in the title may be replaced by its non-abbreviated name. 

Response of Authors: 

“NCAM” has been replaced by “Neural Cell Adhesion Molecule” in the revised manuscript.

  1. Refs: Be consistent for writing the authors’ names.  e.g.  ref 18. Zhou GP, …  to Zhou G. P. ?; ref 21. Liao SM, … to Liao S. M. 

Response of Authors: 

The all authors’ names in the REFERENCES have been corrected to be consistent in writing format. e.g. ref.18 and ref.21 in the revised manuscript:

  1. Zhou, G.P., Liao, S.M., Chen, D., Huang, R.B. "The Cooperative Effect between Polybasic Region (PBR) and Polysialyltransferase Domain (PSTD) within Tumor-Target Polysialyltransferase ST8SiaII". Curr Top Med Chem., 2019,19(31):2831-2841.

  1. Liao, S. M., Lu, B., Liu, X.H., et al. "Molecular Interactions of the Polysialyltransferase Domain (PSTD) in ST8Sia IV with CMP-Sialic Acid and Polysialic Acid Required for Polysialylation of the Neural Cell Adhesion Molecule Proteins: An NMR Study".Int J Mol Sci., 2020, 21(5):1590. doi:10.3390/ijms21051590.
  2. Figures: More detailed figure legends may be provided, e.g. how the images were generated, what were the residues with black background, etc. 

Response of Authors: 

Many thanks to Reviewer1 for pointing out the big defect in our legends. Now, all defects in the legends have been fixed in the revised manuscript.

Reviewer 2 Report

This manuscript summarizes polysialylation in mammals. Polysialytransferase IV is believed to play some major role in transferring poly-sugars to its acceptor molecule. This process appears to have some medical importance, therefore, understanding how the enzyme works is critical toward drug development. Based on partial NMR data in combination of available structure analysis programs, the authors is able to better predict the interaction of active site residues in polysialytransferase with its substrates. Overall, the manuscript should be informative to its filed.   

suggestions: 

Full name should be given before abbreviation (such as SML, SML, TMD, HBV, etc.)

Author Response

Reviewer 2                                                                 Comments and Suggestions for Authors

This manuscript summarizes polysialylation in mammals. Polysialytransferase IV is believed to play some major role in transferring poly-sugars to its acceptor molecule. This process appears to have some medical importance, therefore, understanding how the enzyme works is critical toward drug development. Based on partial NMR data in combination of available structure analysis programs, the authors are able to better predict the interaction of active site residues in polysialytransferase with its substrates. Overall, the manuscript should be informative to its filed.   

Suggestions: Full name should be given before abbreviation (such as SML, SML, TMD, HBV, etc.)

 Response of Authors:

Many thanks to Reviwer2’s very valuable comments. All full names have been given before abbreviations such as SML, TMD, HBV, etc according to the reviewer’s nice suggestions.

Reviewer 3 Report

The submission from Dr. Zhou is focused on the implementation of Chou’s wenxiang diagrams in the process of neural cell adhesion molecule (NCAM) polysialylation, a feature associated on tumor cell migration and progression.

The authors implement the helical wenxiang diagrams to represent in 2D the a-helixes and predict the which residues can play important roles in the intra- and intermolecular interactions. And apply them in the interaction of the two domains of mammalian polysialyltransferases, polybasic region (PBR) and polysialyltransferase domain (PSTD), and the interaction between the PBR and NCAM’s first fibronectin type III repeat (FN1).

However, the manuscript does not contain new experimental data, only the implementation of the wenxiang diagram in the systems described above, where the important residues involved in these interactions have already been described in the respective publications based on experimental data. In addition, Dr. Zhou in Journal of Theoretical Biology 284 (2011) 142–148, implements the same Wenxiang diagrams also to study protein-protein interactions, in this case based on the hydrophobicity of the amino acids, while here based on the acidity of the residues.

In summary, the manuscript is poor in novelty the data are not new, the authors just apply the wenxiang diagram to the systems already studied, I was hoping that after explaining that the wenxiang diagram can be used to predict this type of interaction, they would apply it in a new system and predict for the first time an important protein-protein interaction. Something that doesn't happen.

Author Response

Reviewer 3                                                                  Comments and Suggestions for Authors

The submission from Dr. Zhou is focused on the implementation of Chou’s wenxiang diagrams in the process of neural cell adhesion molecule (NCAM) polysialylation, a feature associated on tumor cell migration and progression.

The authors implement the helical wenxiang diagrams to represent in 2D the a-helixes and predict the which residues can play important roles in the intra- and intermolecular interactions. And apply them in the interaction of the two domains of mammalian polysialyltransferases, polybasic region (PBR) and polysialyltransferase domain (PSTD), and the interaction between the PBR and NCAM’s first fibronectin type III repeat (FN1).

However, the manuscript does not contain new experimental data, only the implementation of the wenxiang diagram in the systems described above, where the important residues involved in these interactions have already been described in the respective publications based on experimental data. In addition, Dr. Zhou in Journal of Theoretical Biology 284 (2011) 142–148, implements the same Wenxiang diagrams also to study protein-protein interactions, in this case based on the hydrophobicity of the amino acids, while here based on the acidity of the residues.

In summary, the manuscript is poor in novelty the data are not new, the authors just apply the wenxiang diagram to the systems already studied, I was hoping that after explaining that the wenxiang diagram can be used to predict this type of interaction, they would apply it in a new system and predict for the first time an important protein-protein interaction. Something that doesn't happen.

Response of Authors:

No, this manuscript displays a novelty graphic system in studying protein-protein/ligand interactions incorporating wenxiang diagrams, molecular modeling and NMR spectroscopy. The reasons are as follows:

  1. As the Reviewer3 said that the wenxiang diagrams we published in Journal of Theoretical Biology 2011, 284, 142–148, perfectly predicted the dimer-dimer protein interactions; The wenxiang diagrams we published inCurrent Topics in Medicinal Chemistry, 2013, 13, 1152-1163, determined the key amino acids which resulted in instability of the helix, and further proposed a prion protein misfolding mechanism.

In this manuscript, however, we found two intermolecular and one intramolecular interaction could be perfectly integrated in a graphic system, and each interaction could be affected by another interaction. Therefore, this is the first time to build a novel graphic system for analyzing NCAM polysialylation mechanism.

  1. In addition, Reviewer3 proposed that “I was hoping that after explaining that the wenxiang diagram can be used to predict this type of interaction, they would apply it in a new system and predict for the first time an important protein-protein interaction. Something that doesn't happen”.

            PSTD and PBR belong to same one molecule ST8Sia IV, and so far, there is no experimental data to          prove the interaction between PSTD and PBR. Therefore, our combined wenxiang diagrams of PSTD- PBR have first time predicted the interaction between them, and the crucial amino acids are A263 and        Y267 from the PSTD, and F88 and L89 from the PBR.  Above predicting results should reasonable due            to the intuitive, and clear logic of the integrated graphic system. Actually, these predictions have been       supported by the recent molecular modeling analysis.

We very appreciate Reviewer3’s comments, which help us to add the following sentences to COCLUSION for further improvement in our revised manuscript:

“In summary, our combined wenxiang diagrams of PSTD-PBR interaction are first time for predicting the PSTD-PBR intramolecular interaction, and the crucial amino acids from these two domains.   These predictions should be reasonable due to intuitive and clear logic display of the integrated graphic system, and have been supported by recent molecular modeling analysis. We found that two intermolecular and one intramolecular interaction could be perfectly integrated in a graphic system, and each interaction is able to be affected by another interaction. As shown in Fig. 4, one can images that the PBR’s conformation should be changed when the PBR is bound to FN1, and induces a conformational change of the PSTD, then results in the polySia’s release from the bound PSTD, and finally the free polySia would be bound to NCAM. Thus, this novel graphic system helps us to deeply understand NCAM polysialylation mechanism, as well as drug design of inhibiting tumor cell migration”.

Round 2

Reviewer 3 Report

Thank you for the answers and explanation. The novelty of the work became clearer to me. Therefore, I recommend the manuscript for publication.